# Epidemiology of reported serious adverse drug reactions due to anti-infectives using nationwide database of Thailand

Sopit Sittiphan[1,2], Apiradee Lim[1]*, Haris Khurram[3,1]*, Nurin Dureh[1], Kwankamon Dittakan[4]

1 Department of Mathematics and Computer Science, Faculty of Science and Technology, Prince of Songkla University, Pattani Campus, Pattani, Thailand, 2 Department of Health Consumer Protection and Public Health Pharmacy, Narathiwat Provincial Public Health Office, Narathiwat, Thailand, 3 Department of Sciences and Humanities, National University of Computer and Emerging Sciences, Chiniot-Faisalabad Campus, Chiniot, Pakistan, 4 College of Computing, Prince of Songkla University, Phuket campus, Phuket, Thailand

* apiradee.s@psu.ac.th (AL); hariskhurram2@gmail.com (HK)

## Abstract

Serious Adverse Drug Reactions (ADRs) can cause a longer stay, which can result in fatal outcomes. Understanding the prognostic factors for the serious ADRs play a vital role in developing appropriate serious ADR prevention strategies. This study aimed to analyze nationwide database in Thailand to identify predisposing factors associated with the serious ADRs, explore drug exposure, distribution of serious ADRs, types of ADRs, and classify the determinants of serious ADR due to anti-infective in Thailand. The national database of anti-infective-induced ADRs from January 2012 to December 2021 in Thailand's 77 provinces, Thai Vigibase at the Health Product Vigilance Center (HPVC), was considered. After pre-processing, frequencies and percentages were used to investigate the distribution of ADR seriousness. To determine the significance of the independent variables on the seriousness of anti-infective-induced ADRs, logistic regression and the Classification and Regression Tree (CART) model were performed. A p-value < 0.05 was considered statistically significant. A total of 82,333 ADR cases, of which 20,692 were serious ADRs (25.13%). Serious ADRs is statistically associated with region, gender, ethnicity, age, type of patient, history of drug allergy, chronic disease and dose frequency (p-value < 0.001). The most commonly reported serious ADRs were in the South region of Thailand (OR = 1.92, 95% CI = 1.88–1.97), followed by the North region (OR = 1.68, 95% CI = 1.64–1.71) of Thailand. Gender and history of drug allergy were also statistically associated with the seriousness of ADRs (p-value = 0.001). Reported ADRs revealed that patients were males (OR = 1.11, 95% CI = 1.11–1.13) and those with a prior history of drug allergy (OR = 1.22, 95% CI = 1.20–1.24) were more likely to experience serious ADRs. The risk of having an ADR reported as serious was significantly higher in patients aged 60 and over (OR = 1.42, 95% CI = 1.39–1.46) and patients aged 40–59 years (OR = 1.34, 95% CI = 1.31–1.37) compared to patients aged 0–19 years. IPD patients most commonly associated with serious ADRs. The results of this study will enable healthcare professionals to use caution when prescribing to those groups. Furthermore, developing a reporting system to reduce

**Data availability statement:** The data that support the findings of this study are available as Supporting information.

**Funding:** The author(s) received no specific funding for this work.

**Competing interests:** The authors have declared that no competing interests exist.

serious ADR evidence, such as software with electronic prescribing databases or applications that enable efficient detection of ADRs in high-risk groups, was critical in order to closely monitor and improve patient safety.

## Introduction

Adverse Drug Reactions (ADRs), which can occur in any drug class and cause global morbidity and hospitalization, will be a public health issue. [1–4]. According to the study's findings, the ADR admissions incidence ranged from 0.60% to 7.0%, with the median length of hospital stay varying from 3 to 8.7 days [5]. ADRs cause not only unnecessary morbidity but also mortality. According to other studies, fatal adverse drug reactions accounted for 3% of all deaths in the population [6], and more than 80% of reported drug-drug interactions-adverse drug reactions (DDI- ADRs) were serious, with 7% being fatal [7]. Serious ADRs can cause a longer stay, which can result in fatal outcomes [8]. A serious ADR is defined as any event or reaction that results in death or life threatening, requires hospital admission or prolongation of an existing hospital stay, result in persistent or significant disability/incapacity, or is cancers, congenital anomalies, birth defects, or other medically important conditions [9].

Anti-infective drugs, including antibiotics, antifungals, antivirals, and antiprotozoals, have saved countless lives and alleviated suffering [10,11]. However, anti-infective drugs are the leading cause of adverse drug reactions (ADRs) worldwide [6,12–15]. In particular, antibiotics are the most common cause of life-threatening off-target immune-mediated drug reactions, including severe cutaneous adverse reactions [16]. Antibiotics are the most prescribed drugs worldwide and are growing in use. Moreover, antibiotic overuse increases ADRs and antimicrobial resistance [17,18]. The study in Saudi Arabia also found that vancomycin and ceftriaxone, which are anti-infective drugs, were the most commonly associated with serious adverse drug reactions [15].

Understanding the prognostic factors for the seriousness of adverse drug reactions due to anti-infectives plays a vital role in developing appropriate serious ADR prevention strategies [8] and improving the survival rate of patients. The majority of studies have focused on factors influencing the occurrence of ADRs [19–26]. However, only a few studies explore the related risk factors for the seriousness of ADRs [8,27,28] and analyze data in Thailand [29]. Thai Vigibase, the HPVC-regulated national spontaneous reporting database, began in 1984. Health professionals and marketing authorization holders in the public and private sectors must report nationwide ADRs [30]. The previous research [29] used Thai Vigibase to study the factors associated with serious outcomes of ADRs caused only by Dimenhydrinate, which are of limited scope. Therefore, the purpose of this study was to analyze a large-scale nationwide database in Thailand to investigate the predisposing factors associated with serious ADRs and explore drug exposure to ADRs and their pattern. Also, identify the distribution of serious ADRs. Moreover, explores the types of ADRs, and classifies the determinants of serious ADRs due to anti-infectives in Thailand.

## Materials and methods

### Data source and study population

The Health Product Vigilance Center's retrospective anti-infective-induced ADR data from Thailand's 77 provinces from January 2012 to December 2021 were utilized. The Thai Food and Drug Administration (FDA) approved database access with patient anonymity. This study included a total of 82,333 ADR cases.

## Inclusion and exclusion criteria

The study included reported cases with at least one anti-infective ADR case report as suspected, causality assessment of ADRs using Naranjo's algorithm, WHO's criteria, and Thai algorithm, and anti-infective-induced ADR reports classified as "certain", "probable", or "possible". The study excluded reports that did not identify the seriousness of ADRs, drugs, or were classified as "unlikely". Furthermore, cases identified as non-anti-infective drugs were excluded from the analysis.

## Data description and pre-processing

The data include patient characteristics (gender, age, ethnicity, type of patient, history of drug allergy, and chronic disease), medication records (disease, ATC class, dosage form, dose frequency, anti-infective drug group), and adverse event information (the seriousness of ADRs, onset of ADRs, province, and response to ADRs). We divided the participants into four age groups: 0 to 19, 20 to 39, 40 to 59, and 60 and older. We separated the participants' ethnicities into two categories: Thai and non-Thai. We classified the type of patient into two groups: inpatient (IPD) and outpatient (OPD). We classified the history of drug allergy as either yes or no. Moreover, diabetes mellitus (DM), hypertension (HT), dyslipidemia, ischemic heart disease, hepatic function abnormalities, and renal efficiency are among the various chronic diseases.

Diseases or medical conditions were classified by the International Classification of Diseases and Related Health Problems 10th Revision (ICD-10) list by the World Health Organization (WHO) [31]. Anatomical Therapeutic Chemical (ATC) classified drugs into 14 classes, including alimentary tract and metabolism, blood and blood-forming organs, cardiovascular system, dermatological, genito-urinary system and sex hormones, systemic hormonal preparations (excluding insulin and sex hormones), anti-infective agents for systemic use, antineoplastic and immunomodulating agents, musculo-skeletal system, nervous system, antiparasitic products, insecticides and repellents, respiratory system, sensory organs, and various [32]. The following dosage forms were classified based on the route of administration: oral, topical, rectal, parenteral, vaginal, inhaled, liquid ophthalmic, and others. The dose frequency was divided into three categories: less than 2 times, between 2 and 5 times, and greater than 5 times. Furthermore, anti-infective drugs are classified based on the function of anti-infective agents such as anti-mycobacterial, antifungal, antiviral, antimalarial, anthelmintic, anti-tuberculosis, and antibiotic [33]. Antibiotics were sub-grouped by chemical structure.

The seriousness of ADRs is divided into serious ADRs and non-serious ADRs. Serious ADR is defined as an adverse drug reaction that caused any of the following six conditions to the patient: 1) Death, 2) A life-threatening situation, 3) Hospitalization, 4) Persistent or significant disability/incapacity, 5) Congenital anomaly/birth defect, and 6) A medically significant situation [19]. The onset of ADRs was analyzed from the first dose of the investigational drug to the first occurrence of the ADRs of interest, classified as < 5 days or > 5 days [34]. We divided Thailand's 77 provinces where ADRs occurred into five regions: Central, East, North, North-East, and South [35]. Furthermore, responses to ADRs include redosing but changed route administration; redosing but dose increased; redosing but dose reduced; redosing and dose not changed, and stopping using the product.

## Statistical analysis

Frequency distribution and cross-tabulation were used to examine the factor, exposure, and distribution of the seriousness of anti-infective-induced ADRs. Pearson Chi-square test was used to evaluate the univariate association between different factors and types of ADR. The

logistic regression model was used to investigate the factors and their role in serious ADRs. Finally, to classify the determinants of serious anti-infective-induced ADRs, the Classification and Regression Tree (CART) model was used. A p-value < 0.05 was considered as statistically significant.

## Ethical approval

This is an observational study based on secondary data and approved by the Research Ethics Committee for Science, Technology and Health Science, Prince of Songkla University, Thailand (number: psu.pn.1-004/66).

## Results

A total of 82,333 ADR cases, of which 20,692 were serious ADRs (25.13%). The seriousness of ADRs is statistically associated with region, gender, ethnicity, age, type of patient, history of drug allergy, chronic disease; diabetes mellitus (DM), hypertension (HT), dyslipidemia, ischaemic heart disease, hepatic function abnormalities, renal insufficiency, and dose frequency (p-value < 0.001). The most commonly reported serious ADRs were in the North-East region of Thailand (29.3%), followed by Thailand's North region (27.8%), South region (25.9%), East region (23.5%), and Central region (20.3%), respectively. The female-male rate of serious ADRs was 1.12. 24.9% of Thai patients reported serious ADRs. The proportion of serious ADRs in elderly patients (60 years and over 60 years) 28.8% was higher than the serious ADRs in adult patients. Overall, these serious ADRs were more common in IPD patients than OPD patients. A total of 27.1% of serious cases reported had a prior history of drug allergy. Furthermore, most serious ADR cases had chronic disease including, diabetes millitus (DM), hypertension (HT), dyslipidemia, ischaemic heart disease, hepatic function abnormalities, and renal insufficiency (Table 1).

Table 2 presents a list of commonly occurring serious ADRs within the Anatomical Therapeutic Drug Class (ATC) and anti-infective drug category. The ATC classes were alimentary tract and metabolism (44.0%), followed by dermatological (39.1%) and antiparasidic products (36.4%), respectively. Antibiotics categorized by chemical structure were sulfone (66.7%), followed by betalactam (57.9%), and polymyxin (55.8%), respectively.

Moreover, the anatomical therapeutic drug class (ATC) and anti-infective drug type are statistically associated with the seriousness of ADRs (p-value < 0.001). When classified by disease classification and ADRs classified by system organ class (adverse disorder), are statistically associated with a p-value < 0.001. Anti-infective use primarily caused certain infections and parasitic diseases (32.8%), followed by disease of the nervous system (31.4%) and symptoms, signs, and abnormal clinical and laboratory findings not elsewhere classified (27.3%), respectively. Infections and infestations (78.4%) were the most adverse disorders, followed by injury, poisoning, and procedural complications (74.0%) and vascular disorders (67.1%), respectively (Table 3).

Fig 1 shows the type of anti-infective ADR seriousness. 74.87% of all ADRs were found to be non-serious. Despite 25.13% being serious ADRs. That is, 21.84% was hospitalization, followed by others (3.02%), 0.24 was death, and 0.03% was disability, respectively.

The most responses to non-serious ADRs were reducing and dose increasing (100%), followed by reducing but changing route administration (83.2%) and redosing but dose not changing (80.5%), respectively. On the other hand, the most common responses to serious ADRs were redosing but reducing the dose (44.8%), followed by stopping the drug (25.3%) and redosing but not changing the dose (19.5%), respectively as Fig 2.

When non-serious ADRs occurred, all responses were used, such as, stop using the drug, redosing and does not change, redosing but dose increased, redosing but dose reduced, and

**Table 1. Distribution and association of demographic characteristics's patients, dose frequency and ADRs onset with the seriousness of anti-infective-induced-ADRs; N = 82,333.**

| Variable | Categories | Non-serious N (%) | Serious N (%) | p-value |
|---|---|---|---|---|
| Region | Central | 19,075 (79.8%) | 4,832 (20.2%) | < 0.001 |
| | East | 7,374 (76.5%) | 2,264 (23.5%) | |
| | North | 9,785 (72.2%) | 3,762 (27.8%) | |
| | North East | 15,068 (70.7%) | 6,232 (29.3%) | |
| | South | 10,075 (74.1%) | 3,516 (25.9%) | |
| | Missing | 264 (75.4%) | 86 (24.6%) | |
| Gender | Male | 25,002 (72.0%) | 9,747 (28.0%) | < 0.001 |
| | Female | 36,452 (77.0%) | 10,873 (23.0%) | |
| | Missing | 187 (72.2%) | 72 (27.8%) | |
| Ethnicity | Thai | 58,083 (75.1%) | 19,227 (24.9%) | 0.031 |
| | Other | 856 (78.0%) | 242 (22.0%) | |
| | Missing | 2,702 (68.8%) | 1,223 (31.2%) | |
| Age (year) | 0–19 | 9,446 (80.0%) | 2,365 (20.0%) | < 0.001 |
| | 20–39 | 23,677 (76.0%) | 7,463 (24.0%) | |
| | 40–59 | 16,052 (73.4%) | 5,826 (26.6%) | |
| | 60 and above | 12,466 (71.2%) | 5,038 (28.8%) | |
| Type of patient | IPD | 26,225 (65.9%) | 13,595 (34.1%) | < 0.001 |
| | OPD | 29,166 (84.2%) | 5,472 (15.8%) | |
| | Missing | 6,250 (79.4%) | 1,625 (20.6%) | |
| History of Drug Allergy | No | 44,716 (75.7%) | 14,321 (24.3%) | < 0.001 |
| | Yes | 8,795 (72.9%) | 3,273 (27.1%) | |
| | Missing | 8,130 (72.4%) | 3,098 (27.6%) | |
| Chronic disease | No | 59,071 (75.1%) | 19,629 (24.9%) | < 0.001 |
| | Yes | 2,570 (70.7%) | 1,063 (29.3%) | |
| Diabetes millitus (DM) | No | 60,605 (75.0%) | 20,194 (25.0%) | < 0.001 |
| | Yes | 1,036 (67.5%) | 498 (32.5%) | |
| Hypertension (HT) | No | 59,966 (75.0%) | 19,978 (25.0%) | < 0.001 |
| | Yes | 1,675 (70.1%) | 714 (29.9%) | |
| Dyslipidemia | No | 60,892 (74.9%) | 20,359 (25.1%) | < 0.001 |
| | Yes | 749 (69.2%) | 333 (30.8%) | |
| Ischaemic Heart Disease | No | 61,483 (74.9%) | 20,620 (25.1%) | 0.031 |
| | Yes | 158 (68.7%) | 72 (31.3%) | |
| Hepatic Function Abnomalities | No | 61,547 (74.9%) | 20,629 (25.1%) | < 0.001 |
| | Yes | 94 (59.9%) | 63 (40.1%) | |
| Renal Insufficiency | No | 61,245 (75.0%) | 20,428 (25.0%) | < 0.001 |
| | Yes | 396 (60.0%) | 264 (40.0%) | |
| | Missing | 25,569 (72.9%) | 9,495 (27.1%) | |
| Dose Frequency | < 2 | 31,251 (72.8%) | 11,653 (27.2%) | < 0.001 |
| | > 5 | 14,642 (75.1%) | 4,859 (24.9%) | |
| | 2 to 5 | 15,748 (79.0%) | 4,180 (21.0%) | |
| ADRs Onset (day) | < 5 days | 24,641 (75.1%) | 8,167 (24.9%) | 0.199 |
| | > 5 days | 37,000 (74.7%) | 12,525 (25.3%) | |

redosing but changed route administration. Conversely, a different approach was taken when serious ADRs emerged. Redosing but dose increased was not used for all serious cases, stop using the drug that was used for all serious ADRs. Particularly in cases of disability and death (Fig 3).

**Table 2. Distribution and association of the Anatomical Therapeutic Drug Class (ATC) and anti-infective drug type with the seriousness of anti-infective-induced-ADRs; N = 82,333.**

| Variable | Categories | Non-serious N(%) | Serious N(%) | p-value |
|---|---|---|---|---|
| The Anatomical Therapeutic Drug Class (ATC) | Alimentary tract and metabolism(a) | 163 (56.0%) | 128 (44.0%) | < 0.001 |
| | Antiparasitic products(p) | 35 (63.6%) | 20 (36.4%) | |
| | Dermatologicals(d) | 14 (60.9%) | 9 (39.1%) | |
| | General antiinfectives, systemic(j) | 60,766 (74.9%) | 20,402 (25.1%) | |
| | Genito urinary system and sex hormones(g) | 629 (83.3%) | 126 (16.7%) | |
| | Sensory organs(s) | 24 (88.9%) | 3 (11.1%) | |
| Anti-Infective Drug Type | Anthelmintics | 4 (57.1) | 3 (42.9) | < 0.001 |
| | Antibiotic-Aminoglycoside | 453 (65.4) | 240 (34.6) | |
| | Antibiotic-Aminosalicylic acid | 2 (100) | 0 (0) | |
| | Antibiotic-Betalactam | 858 (42.1) | 1,182 (57.9) | |
| | Antibiotic-Carbapenem | 1,528 (72.6) | 576 (27.4) | |
| | Antibiotic-Cephalosporin | 18,391 (79.8) | 4,652 (20.2) | |
| | Antibiotic-Chloramphenicol | 59 (93.7) | 4 (6.3) | |
| | Antibiotic-Fluoroquinolone | 6,193 (80) | 1,551 (20) | |
| | Antibiotic-Fusidic acid | 8 (66.7) | 4 (33.3) | |
| | Antibiotic-Glycopeptides | 122 (60.1) | 81 (39.9) | |
| | Antibiotic-Glycylcyclines | 15 (93.8) | 1 (6.2) | |
| | Antibiotic-Lincosamide | 5,112 (82.7) | 1,073 (17.3) | |
| | Antibiotic-Macrolide | 1,602 (79.8) | 405 (20.2) | |
| | Antibiotic-Monoxycarbolic acid | 13 (86.7) | 2 (13.3) | |
| | Antibiotic-Nitrofuran | 7 (53.8) | 6 (46.2) | |
| | Antibiotic-Nitroimidazole | 621 (84.5) | 114 (15.5) | |
| | Antibiotic-Oxazolidinone | 11 (45.8) | 13 (54.2) | |
| | Antibiotic-Para Aminosalicylic acid (PAS) | 10 (100) | 0 (0) | |
| | Antibiotic-Penicillin | 14,735 (76.5) | 4,516 (23.5) | |
| | Antibiotic-Phosphonic acid | 141 (77) | 42 (23) | |
| | Antibiotic-Polymyxin | 65 (44.2) | 82 (55.8) | |
| | Antibiotic-Sulfonamide | 3,180 (63.7) | 1,814 (36.3) | |
| | Antibiotic-Sulfone | 84 (33.3) | 168 (66.7) | |
| | Antibiotic-Tetracycline | 2,144 (78.4) | 592 (21.6) | |
| | Antifungal | 526 (63.7) | 300 (36.3) | |
| | Antimalarial | 31 (64.6) | 17 (35.4) | |
| | Antiviral | 4,115 (73.2) | 1,504 (26.8) | |
| | Anti-Tuberculosis | 1,601 (47.8) | 1,746 (52.2) | |

Table 4 presents risk factors associated with the seriousness of anti-infective-induced ADRs, after adjusting for potential confounding variables. The region was statistically associated with the seriousness of ADRs (p-value < 0.001). The most commonly reported serious ADRs were in the South region of Thailand (OR = 1.92, 95% CI = 1.88–1.97), followed by the North region (OR = 1.68, 95% CI = 1.64–1.71), the North-East region (OR = 1.58, 95% CI = 1.55–1.61), and the East region (OR = 1.17, 95% CI = 1.13–1.20), compared to the Central region of Thailand. Gender and history of drug allergy were also statistically associated with the seriousness of ADRs (p-value = 0.001). From the reported ADRs, males (OR = 1.11, 95% CI = 1.11–1.13) and those with a prior history of drug allergy (OR = 1.22, 95% CI = 1.20–1.24) experienced more serious ADRs compared to females and those without a history of drug allergy, respectively. The risk of having an

**Table 3. Distribution and association of diseases classification and ADRs classified by system organ class (Adverse Disorder) with the seriousness of anti-infective-induced-ADRs; N = 82,333.**

| Variable | Categories | Non-serious N(%) | Serious N(%) | p-value |
|---|---|---|---|---|
| 2024 ICD-10-CM Diseases classification | Certain conditions originating in the perinatal period | 102 (85.0%) | 18 (15.0%) | < 0.001 |
| | Certain infectious and parasitic diseases | 14,891 (67.2%) | 7,281 (32.8%) | |
| | Congenital malformations, deformations and chromosomal abnormalities | 47 (81.0%) | 11 (19.0%) | |
| | Disease of Thai Traditional medicine | 171 (76.7%) | 52 (23.3%) | |
| | Diseases of the circulatory system | 745 (76.7%) | 226 (23.3%) | |
| | Diseases of the digestive system | 3,239 (81.8%) | 721 (18.2%) | |
| | Diseases of the eye and adnexa | 933 (84.7%) | 169 (15.3%) | |
| | Diseases of the genitourinary system | 5,807 (81.5%) | 1,314 (18.5%) | |
| | Diseases of the musculoskeletal system and connective tissue | 1,291 (79.5%) | 333 (20.5%) | |
| | Diseases of the nervous system | 251 (68.6%) | 115 (31.4%) | |
| | Diseases of the respiratory system | 8,018 (78.0%) | 2,258 (22.0%) | |
| | Diseases of the skin and subcutaneous tissue | 4,937 (78.0%) | 1,391 (22.0%) | |
| | Endocrine, nutritional and metabolic diseases | 159 (74.6%) | 54 (25.4%) | |
| | External causes of morbidity | 365 (82.8%) | 76 (17.2%) | |
| | Factors influencing health status and contact with health services | 1,031 (81.4%) | 236 (18.6%) | |
| | Injury, poisoning and certain other consequences of external causes | 3,872 (75.3%) | 1,273 (24.7%) | |
| | Mental, Behavioral and Neurodevelopmental disorders | 39 (90.7%) | 4 (9.3%) | |
| | Other specified postsurgical states | 3 (100.0%) | 0 (0%) | |
| | Pregnancy, childbirth and the puerperium | 794 (84.9%) | 141 (15.1%) | |
| | Symptoms, signs and abnormal clinical and laboratory findings, not elsewhere classified | 3,401 (72.7%) | 1,275 (27.3%) | |
| Adverse Disorder | Blood and lymphatic | 437 (53.8%) | 375 (46.2%) | < 0.001 |
| | Cardiac | 1,121 (80.2%) | 276 (19.8%) | |
| | Congenital, familial and genetic | 29 (36.7%) | 50 (63.3%) | |
| | Ear and labyrinth | 29 (72.5%) | 11 (27.5%) | |
| | Endocrine | 64 (90.1%) | 7 (9.9%) | |
| | Eye | 797 (81.1%) | 186 (18.9%) | |
| | Gastrointestinal | 1,680 (85.1%) | 295 (14.9%) | |
| | General disorders and administration | 1,690 (82.8%) | 352 (17.2%) | |
| | Hepatobiliary | 695 (41.2%) | 992 (58.8%) | |
| | Immune | 11,720 (69.5%) | 5,137 (30.5%) | |
| | Infections and infestations | 255 (21.6%) | 926 (78.4%) | |
| | Injury, poisoning and procedural complications | 394 (26.0%) | 1,124 (74.0%) | |
| | Investigations | 235 (67.0%) | 116 (33.0%) | |
| | Metabolism and nutrition disorders | 611 (78.3%) | 169 (21.7%) | |
| | Musculoskeletal and connective tissue | 190 (79.8%) | 48 (20.2%) | |
| | Neoplasms benign, malignant | 3 (33.3%) | 6 (66.7%) | |
| | Nervous system disorders | 503 (81.3%) | 116 (18.7%) | |
| | Pregnancy, puerperium and perinatal | 4 (80.0%) | 1 (20.0%) | |
| | Psychiatric disorders | 72 (78.3%) | 20 (21.7%) | |
| | Renal and urinary disorders | 258 (49.4%) | 264 (50.6%) | |
| | Reproductive system and breast | 88 (80.7%) | 21 (19.3%) | |
| | Respiratory, thoracic and mediastinal | 2,831 (79.9%) | 711 (20.1%) | |
| | Skin and subcutaneous tissue | 36,962 (83.1%) | 7,511 (16.9%) | |
| | Surgical and medical procedures | 5 (83.3%) | 1 (16.7%) | |
| | Vascular disorders | 968 (32.9%) | 1,977 (67.1%) | |

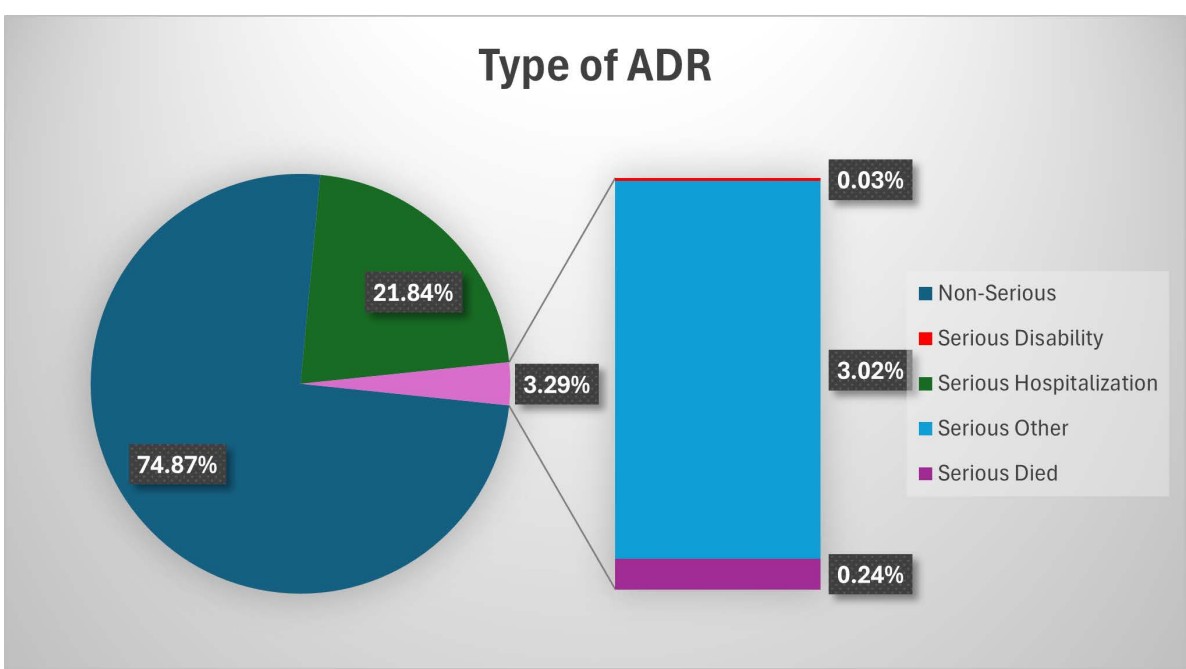

**Fig 1. Type of Anti-Infective-Induced.**

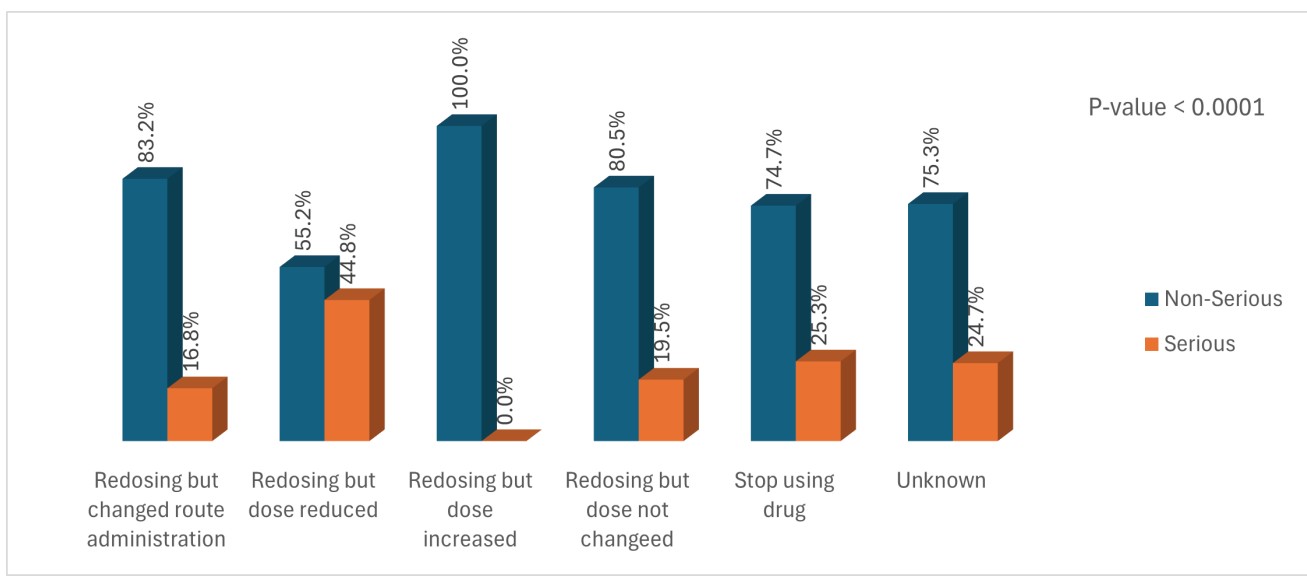

**Fig 2. Serious ADRs in responses to Anti-Infective-Induced ADRs.**

ADR reported as serious was significantly higher in patients aged 60 and over (OR = 1.42, 95% CI = 1.39–1.46) and patients aged 40–59 years (OR = 1.34, 95% CI = 1.31–1.37) compared to patients aged 0–19 years. IPD patients most commonly associated with serious ADRs. Conversely, the risk of having an ADR reported as serious was significantly higher in dermatologists (OR = 4.01, 95% CI = 2.58–6.24) compared to alimentary tract and metabolism. Although anti-infective drug type was significantly associated with the seriousness of ADRs, the impact varied by sub-group.

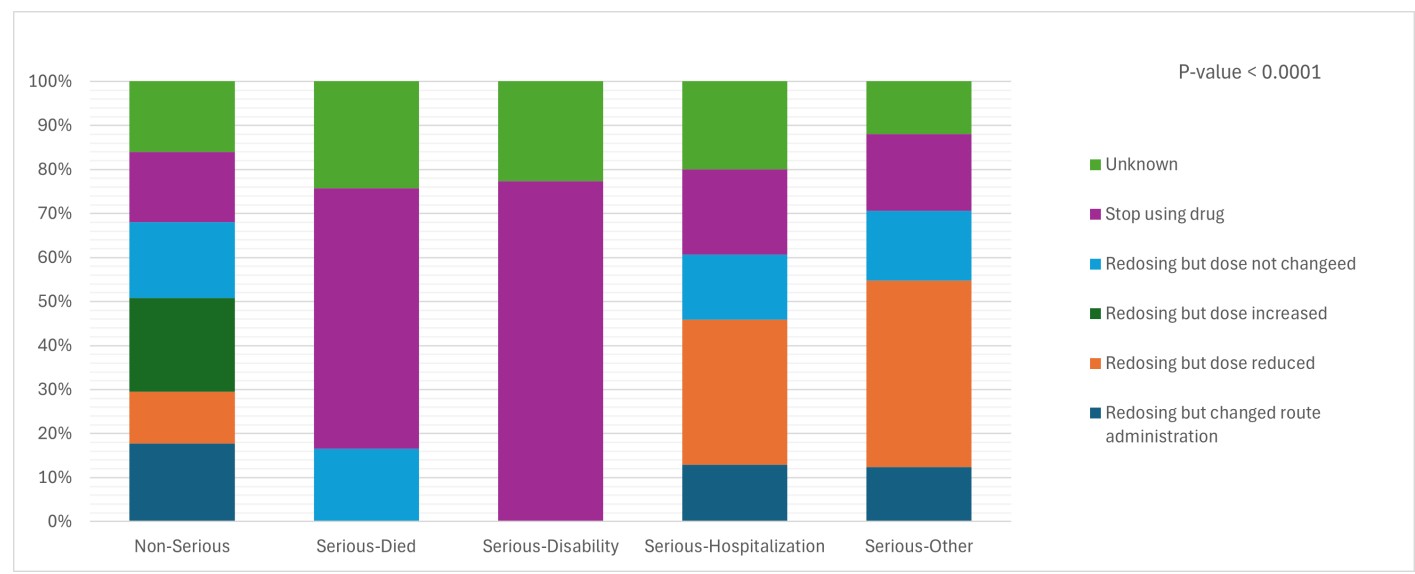

**Fig 3. Association between Redosing and type of Anti-Infective-Induced-ADRs.**

**Table 4. Multiple logistic regression analysis of factors associated with serious adverse drug reactions due to anti-infectives.**

| Variable | Categories | Coefficient⁺ | Crude OR [95% CI] | Adjusted OR [95% CI] |
|---|---|---|---|---|
| Region | Central | Ref. | | |
| | East | 0.15 | 1.21 [1.19–1.24]*** | 1.17 [1.13–1.20]*** |
| | North | 0.52 | 1.52 [1.49–1.54]*** | 1.68 [1.64–1.71]*** |
| | Northeast | 0.46 | 1.63 [1.61–1.66]*** | 1.58 [1.55–1.61]*** |
| | South | 0.65 | 1.38 [1.35–1.40]*** | 1.92 [1.88–1.97]*** |
| Gender | Female | Ref. | | |
| | Male | 0.11 | 1.31 [1.29–1.32]*** | 1.11 [1.10–1.13]*** |
| Ethnicity | Other | Ref. | | |
| | Thai | 0.22 | 1.17 [1.11–1.23]** | 1.25 [1.17–1.33]** |
| Age | 0–19 | Ref. | | |
| | 20–39 | 0.23 | 1.26 [1.24–1.28]*** | 1.26 [1.24–1.29]*** |
| | 40–59 | 0.29 | 1.45 [1.42–1.48]*** | 1.34 [1.31–1.37]*** |
| | 60 and above | 0.35 | 1.61 [1.58–1.65]*** | 1.42 [1.39–1.46]*** |
| Type of Patient | IPD | Ref. | | |
| | OPD | −1.42 | 0.36 [0.36–0.37]*** | 0.24 [0.24–0.25]*** |
| History of Drug Allergy | No | Ref. | | |
| | Yes | 0.20 | 1.16 [1.14–1.18]*** | 1.22 [1.20–1.24]*** |
| Dose Frequency | < 2 | Ref. | | |
| | > 5 | −0.06 | 0.89 [0.88–0.90]*** | 0.94 [0.92–0.96]** |
| | 2 to 5 | −0.31 | 0.71 [0.70–0.72]*** | 0.73 [0.72–0.75]*** |
| ADRs onset | < 5 days | Ref. | | |
| | > 5 days | 0.03 | 1.02 [1.01–1.03] | 1.03 [1.02–1.05] |
| The Anatomical Therapeutic Drug Class (ATC) | Alimentary tract and metabolism(a) | Ref. | | |
| | Antiparasitic products(p) | −0.96 | 0.73 [0.59–0.89] | 0.38 [0.22–0.67] |
| | Dermatologicals(d) | 1.39 | 0.82 [0.61–1.10] | 4.01 [2.58–6.24]** |
| | General antiinfectives, systemic(j) | 0.28 | 0.43 [0.39–0.46]*** | 1.32 [1.06–1.65] |
| | Genito urinary system and sex hormones(g) | 0.73 | 0.26 [0.23–0.28]*** | 2.08 [1.39–3.12] |
| | Sensory organs(s) | −0.01 | 0.16 [0.10–0.24]*** | 0.99 [0.60–1.61] |

*(Continued)*

**Table 4.** (Continued)

| Variable | Categories | Coefficient+ | Crude OR [95% CI] | Adjusted OR [95% CI] |
|---|---|---|---|---|
| Anti-infective Drug Type | Anthelmintics | Ref. | | |
| | Antibiotic-Aminoglycoside | −1.15 | 0.71 [0.42–1.19] | 0.32 [0.29–0.34]*** |
| | Antibiotic-Aminosalicylic acid | −12.21 | – | 0.00 [0.00–9.21E+61] |
| | Antibiotic-Beta lactam | 0.24 | 1.84 [1.10–3.08] | 1.27 [1.22–1.33]*** |
| | Antibiotic-Carbapenem | −1.54 | 0.50 [0.30–0.84] | 0.21 [0.20–0.23]*** |
| | Antibiotic-Cephalosporin | −1.93 | 0.34 [0.20–0.57] | 0.15 [0.14–0.15]*** |
| | Antibiotic-Chloramphenicol | −2.23 | 0.09 [0.05–0.17]*** | 0.11 [0.07–0.16]*** |
| | Antibiotic-Fluoroquinolone | −1.47 | 0.33 [0.20–0.56] | 0.23 [0.22–0.24]*** |
| | Antibiotic-Fusidic acid | −0.44 | 0.67 [0.34–1.29] | 0.65 [0.41–1.02] |
| | Antibiotic-Glycopeptides | −0.68 | 0.89 [0.52–1.50] | 0.51 [0.40–0.64] * |
| | Antibiotic-Glycylcyclines | −13.28 | 0.09 [0.04–0.21] * | 0.00 [0.00–2.73E+20] |
| | Antibiotic-Lincosamide | −1.72 | 0.28 [0.17–0.47] * | 0.18 [0.17–0.19]*** |
| | Antibiotic-Macrolide | −1.21 | 0.34 [0.20–0.57] | 0.30 [0.28–0.31]*** |
| | Antibiotic-Monoxycarbolic acid | −1.71 | 0.21 [0.10–0.42] | 0.18 [0.10–0.34] * |
| | Antibiotic-Nitrofuran | 0.04 | 1.14 [0.60–2.16] | 1.04 [0.57–1.89] |
| | Antibiotic-Nitroimidazole | −2.46 | 0.25 [0.15–0.41] * | 0.09 [0.06–0.12]*** |
| | Antibiotic-Oxazolidinone | −0.82 | 1.58 [0.88–2.83] | 0.44 [0.27–0.71] |
| | Antibiotic-Para Aminosalicylic acid (PAS) | −11.71 | – | – |
| | Antibiotic-Penicillin | −1.04 | 0.41 [0.24–0.68] | 0.35 [0.34–0.36]*** |
| | Antibiotic-Phosphonic acid | −1.85 | 0.40 [0.23–0.67] | 0.16 [0.14–0.18]*** |
| | Antibiotic-Polymyxin | −0.34 | 1.68 [0.99–2.85] | 0.71 [0.59–0.85] |
| | Antibiotic-Sulfonamide | −0.54 | 0.76 [0.45–1.27] | 0.58 [0.56–0.61]*** |
| | Antibiotic-Sulfone | 0.20 | 2.67 [1.58–4.50] | 1.22 [1.09–1.37] |
| | Antibiotic-Tetracycline | −1.11 | 0.37 [0.22–0.62] | 0.33 [0.31–0.34]*** |
| | Antifungal | −0.63 | 0.76 [0.45–1.28] | 0.53 [0.50–0.57]*** |
| | Antimalarial | 0.10 | 0.73 [0.42–1.27] | 1.11 [0.63–1.94] |
| | Antiretroviral | −0.89 | 0.49 [0.29–0.82] | 0.41 [0.39–0.43]*** |
| | AntiTB | 0.00 | 1.45 [0.87–2.44] | 0.00 [0.00–0.00]*** |

+Coefficients were calculated using adjusted model; OR: Odds Ratio; CI: Confidence Interval; * p-value < 0.1; **p-value < 0.05; ***p-value < 0.01.

The antibiotic group had the largest impact of drug type on serious ADRs. The antibiotic group with the highest impact on serious ADRs was betalactam (OR = 1.27, 95% CI = 1.22–1.33). On the other hand, the risk of having an ADR reported as serious was significantly lower in penicillin and sulfonamide compared to the anthelmintics group (OR = 0.35, 95% CI = 0.34–0.36) and (OR = 0.58, 95% CI = 0.56–0.61), whereas, the non-antibiotic group was antiviral (OR = 0.41, 95% CI = 0.39–0.43) and antifungal (OR = 0.53, 95% CI = 0.50–0.57) compared to the anthelmintics group. The seriousness of ADRs was found to be associated with the type of patient, the type of anti-infective drug, and the region, as shown in Fig 4 (the total subjects).

Specifically, ADRs reveal that patients with IPD (Node 1) who received antifungal, anti-TB, or antibiotics like sulfone, betalactam, or monoxycarbolic acid (Node 8), as well as those reported in all five regions of Thailand (Nodes 15 and 16), were most at risk of experiencing serious ADRs. Meanwhile, those who were OPD patients (Node 2) and received anti-TB, anti-malaria, or oxaolidinone (Node 5), also reported in the North-East region of Thailand, were likely to experience serious ADR (Node 12) when compared with ADR reports in other regions (Node 11). Fig 5–6 show the CART model's seriousness in anti-infective-induced ADR prediction.

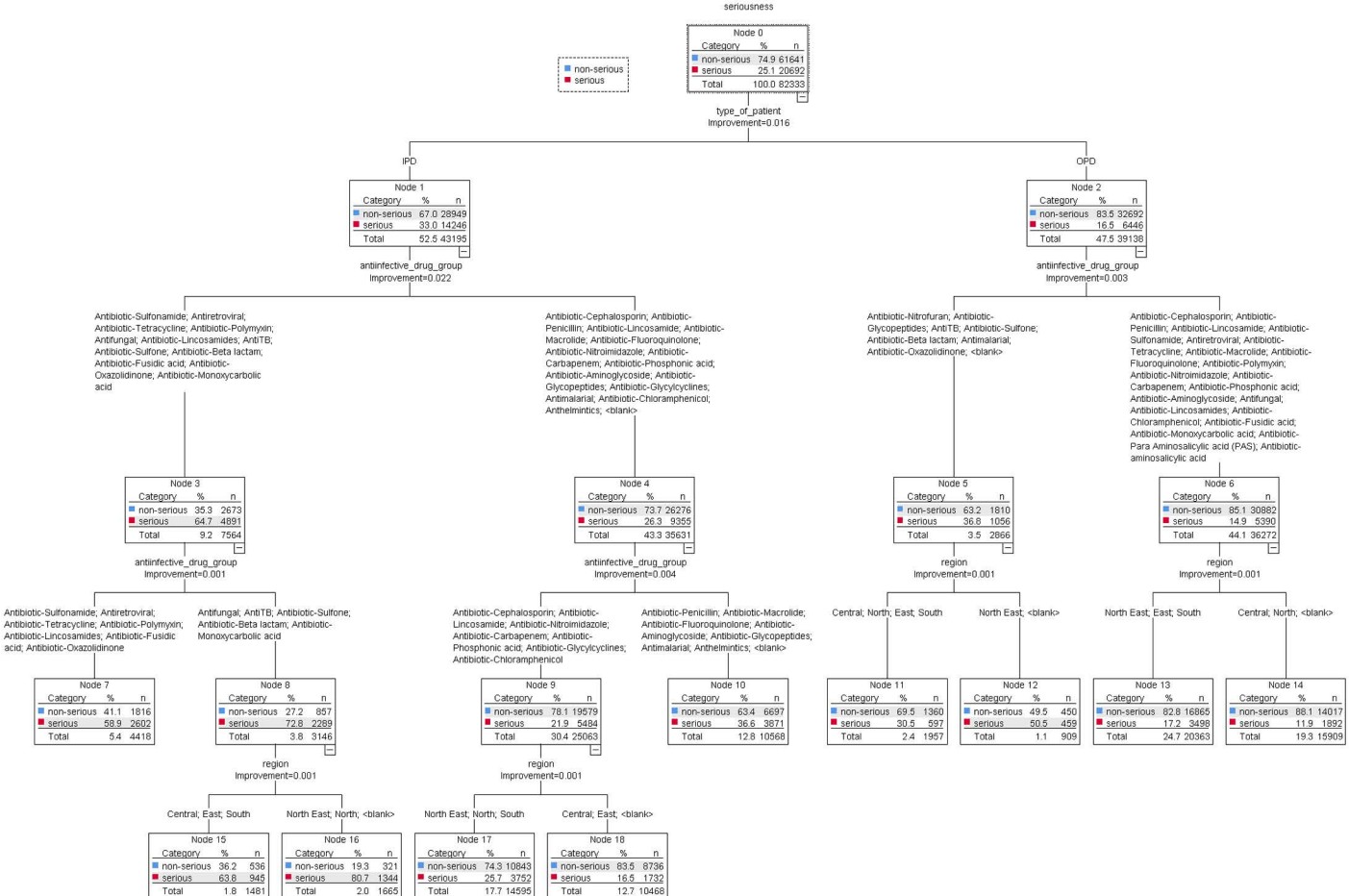

**Fig 4. Classification and regression tree (CART) model to determine the role of anti-infective-induced ADRs in the classification of serious ADRs.**

We found that factors such as patient type, region, dose frequency, age, and gender significantly influence serious ADRs. The CART model in Fig 5 shows that the reported ADRs in IPD patients and males were most commonly associated with serious ADRs. Reported ADRs showed that people with IPD (Node 1), people who lived in the North-East, North, and South regions of Thailand (Node 4), and people who got doses less than twice a week (Node 9) were more likely associated with serious ADRs. Additionally, individuals aged 60 years and above, 20–39 years, and 40–59 years (Node 12), as well as males (Node 14), were more likely to experience serious ADRs compared to their female counterparts (Node 13). Conversely, OPD patients (Node 2) from the North-East, East, and South regions of Thailand reported more serious ADRs (Node 5) than those from the Central and North regions of Thailand (Node 6). The details of the serious classification and the non-serious are shown in Fig 6.

## Discussion

In the present analysis, a total of 82,333 ADR cases, of which 20,692 were serious ADRs (25.13%). The lower rate of serious ADRs may indicate better quality and safety in the patients evaluated or under-reporting of serious ADRs from the pharmacovigilance spontaneous reporting system [36]. A few studies explore the related risk factors for the seriousness

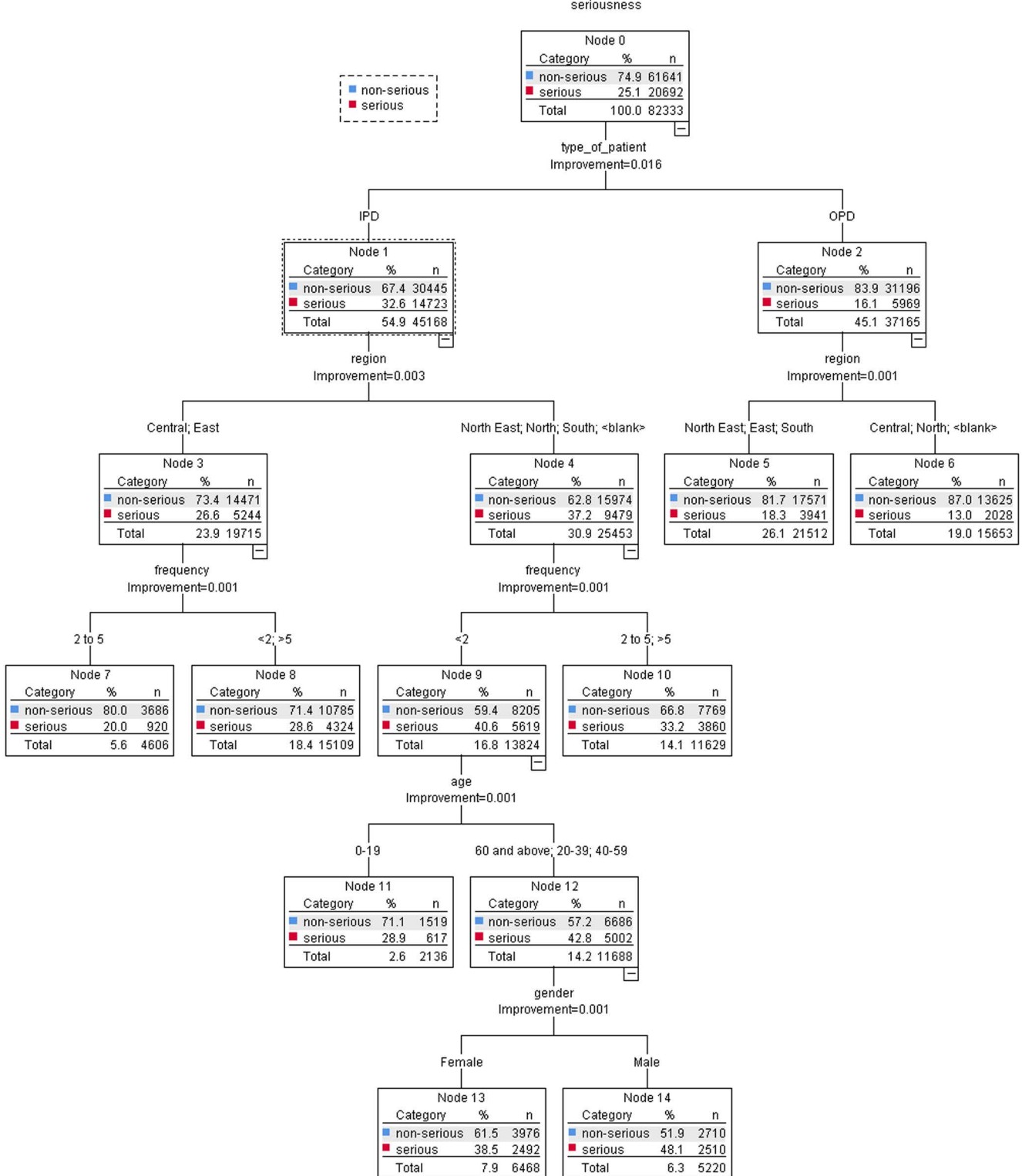

**Fig 5. Classification and Regression Tree (CART) Model to determine the role of demographic factors in the classification of serious ADRs.**

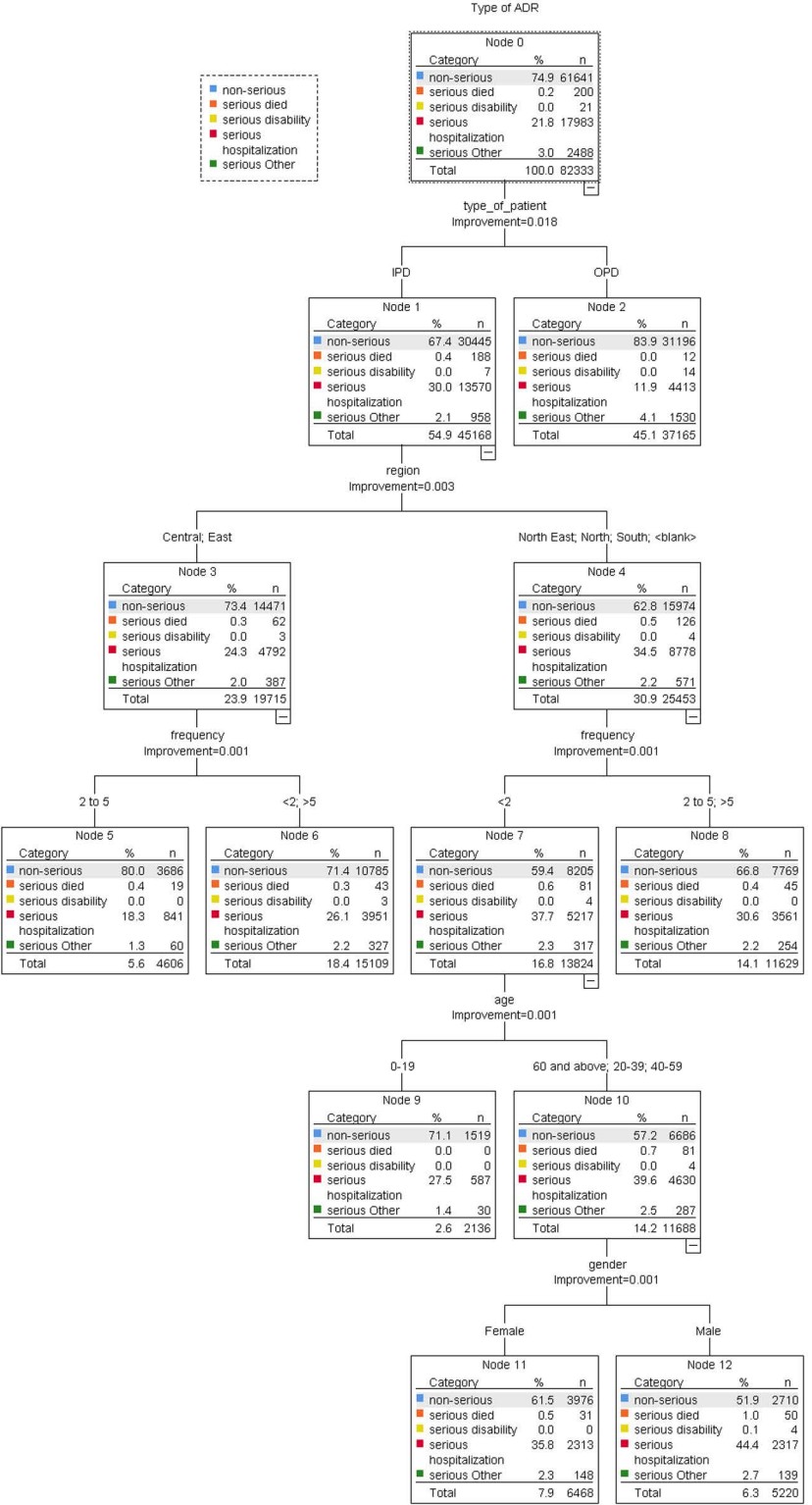

**Fig 6. Classification and Regression Tree (CART) Model to determine the role of demographic factors in the classification of the type of ADR.**

of ADRs. The study conducted in South Korea [8] revealed that polypharmacy and liver function tests (AST/ALT ratio) must be monitored carefully within high-risk groups for serious ADRs. Moreover, the study conducted in China [28] found that age, number of medications and illnesses, level of medical institution, history of adverse reactions, seasons, and type and method of medication were all factors that affected serious ADRs. On the other hand, the present study's finding shows that serious ADRs are statistically associated with region, gender, ethnicity, age, type of patient, history of drug allergy, chronic disease, and dose frequency (p-value < 0.001). This study shows a correlation between the risk of serious ADRs and regions that were statistically associated with the seriousness of ADRs (p-value < 0.001). The most commonly reported serious ADRs were in the south region of Thailand (OR = 1.92, 95% CI = 1.88–1.97) compared to the central region of Thailand. The possible reasons may be due to the longer rainy season in the South of Thailand which causes a higher risk of sickness, which is consistent with the study indicating that season is a factor associated with serious ADRs [28]. The present result shows significant risk factors in the logistic regression analysis, and they were present in the terminal node of the CART model that included gender and history of drug allergy and was also statistically associated with the seriousness of ADRs (p-value = 0.001). ADR reports that patients were males (OR = 1.11, 95% CI = 1.11–1.13) and those with a prior history of drug allergy (OR = 1.22, 95% CI = 1.20–1.24) were more likely to experience serious ADRs compared to females and those without a history of drug allergy, respectively. In contrast with some studies [29], females and those with a prior history of drug allergies are more likely to experience serious ADRs, especially type A adverse drug reactions, and the majority of females weigh less than males [21]. On the other hand, a study [36] revealed that gender did not influence the risk of having an ADR reported as serious. This study has addressed that the risk of having an ADR reported as serious was significantly higher in patients aged 60 and over (OR = 1.42, 95% CI = 1.39–1.46) and patients aged 40–59 years (OR = 1.34, 95% CI = 1.31–1.37) compared to patients aged 0–19 years, which is consistent with several studies [8,27–29,36] indicating that elderly people are at high risk for serious ADRs because of age-related physiological changes and multiple drug regimens for various comorbidities, affecting the pharmacokinetics and pharmacodynamics of many drugs [8,36]. IPD patients most commonly associated with serious ADRs. This finding is in agreement with the result from the CART model shows that ADR reports that IPD patients and males most commonly associated with serious ADRs. The possible reasons may be that the medical characteristics of IPD patients differ from those of OPD patients in terms of disease complications, drug prescriptions, and patient compliance [24]. Furthermore, the risk of having an ADR reported as serious was significantly higher in dermatological (OR = 4.01, 95% CI = 2.58–6.24) compared to alimentary tract and metabolism. The antibiotic group had the largest impact of drug type on serious ADRs. The antibiotic group with the highest impact on serious ADRs was beta-lactam (OR = 1.27, 95% CI = 1.22–1.33), in accordance with the previous studies [8,37,38]. Beta-lactams are the most commonly used antibiotics in clinical practice around the world as first-line treatment for many bacterial infections. However, beta-lactams can cause a variety of hypersensitivity reactions, including anaphylaxis, a life-threatening adverse drug reaction (ADR) [39]. Moreover, reports indicate that these are the primary causes of cutaneous adverse drug reactions, which can vary in severity from mild urticaria and maculopapular exanthema (MPE) to life-threatening severe cutaneous adverse reactions (SCARs) such as Stevens-Johnson syndrome (SJS), toxic epidermal necrolysis (TEN), drug reaction with eosinophilia and systemic symptoms (DRESS), and acute generalized exanthematous pustulosis [16,40].

The present study has several points of strength. Firstly, this study conducts a retrospective analysis to evaluate serious adverse drug reactions (ADRs) caused by anti-infectives over an

extended period in Thailand. Furthermore, this study analyzed a large-scale nationwide database in Thailand using LR and CART analysis methods to investigate risk factors associated with serious ADRs that have an impact on health outcomes and the cost of patient hospital care. Policy makers have the influence and opportunity to use the research evidence to alter or develop effective policies in order to prevent serious ADRs and the health authorities can make decisions more quickly to restrict a drug's use or pay close attention to high risk group for patient safety.

However, this study is not without limitations. The necessary data in the ADR reports, such as common chronic diseases, were typically reported voluntarily, resulting in many missing values in the demographic data. Moreover, some variables, such as variables related to the behavior of the patient (smoking, alcohol intake, etc.) were not available in the database. These parameters increase oxidative stress (OS), as measured by the Oxidative Stress Index (OSI) [19]. Moreover, various laboratory data on renal function and liver function, which are new risk factors for serious ADRs and play a major role in drug metabolism and also number of concomitant drugs [8], could not be included in this study because they were not available in the database. The spontaneous reporting system for ADRs likely contributes to underreporting, which may not be the true representative of population. Furthermore, missing data is a limitation due to retrospective studies. Therefore, a prospective study can be conducted in the future to validate the results.

## Conclusions

Anti-infective drugs are widely used and cause the most commonly reported serious ADRs in males, elderly patients, and those with a history of drug allergy. They have also spread to almost every region of Thailand. The beta-lactam antibiotics subgroup had a higher percentage of reported serious ADRs due to the anti-infective drug. Serious ADRs have an impact on healthcare outcomes and patient care costs, which poses a challenge for the healthcare system. The findings from this study could contribute to the appropriate use of antibiotics in clinical practice, as well as knowledge for better communication between healthcare practitioners in developing countries where antibiotic resistance is a major national public health issue. Furthermore, developing a reporting system to reduce serious ADR evidence, such as software with electronic prescribing databases or applications that enable efficient detection of ADRs in high-risk groups using CART model, was critical in order to closely monitor and improve patient safety.

## Supporting information

**S1 Data Set.  Data set used in this study.**
(CSV)

## Acknowledgments

The authors graciously acknowledge the Health Product Vigilance Center (HPVC), and the Food and Drug Administration of Thailand (Thai FDA) for providing the data.

## Author contributions

**Conceptualization:** Sopit Sittiphan, Apiradee Lim, Haris Khurram.

**Data curation:** Sopit Sittiphan.

**Formal analysis:** Sopit Sittiphan, Apiradee Lim, Haris Khurram.

**Funding acquisition:** Sopit Sittiphan.

**Investigation:** Apiradee Lim, Haris Khurram.

**Methodology:** Apiradee Lim, Haris Khurram.

**Resources:** Sopit Sittiphan, Apiradee Lim.

**Software:** Sopit Sittiphan, Apiradee Lim, Haris Khurram.

**Supervision:** Apiradee Lim, Haris Khurram, Nurin Dureh.

**Validation:** Apiradee Lim, Haris Khurram.

**Visualization:** Apiradee Lim, Haris Khurram.

**Writing – original draft:** Sopit Sittiphan, Apiradee Lim, Nurin Dureh, Kwankamon Dittakan.

**Writing – review & editing:** Apiradee Lim, Haris Khurram, Nurin Dureh, Kwankamon Dittakan.

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
