## [Decision Letter · Decision Letter 0]

15 Nov 2024

PONE-D-24-43299Epidemiology of Serious Adverse Drug Reactions Due to Anti-Infectives in ThailandPLOS ONE

Dear Dr. Khurram,

Thank you for submitting your manuscript to PLOS ONE. After careful consideration, we feel that it has merit but does not fully meet PLOS ONE’s publication criteria as it currently stands. Therefore, we invite you to submit a revised version of the manuscript that addresses the points raised during the review  Please submit your revised manuscript by %DATE_REVISION_DUE. If you will need more time than this to complete your revisions, please reply to this message or contact the journal office at plosone@plos.org . Please include the following items when submitting your revised manuscript:

We look forward to receiving your revised manuscript.

Kind regards,

Obed Kwabena Offe Amponsah, PharmD, Ph.D.

Academic Editor

PLOS ONE

Journal Requirements:

2. Ethics statement only appears at the end of the manuscript:

Your ethics statement should only appear in the Methods section of your manuscript. If your ethics statement is written in any section besides the Methods, please move it to the Methods section and delete it from any other section. Please ensure that your ethics statement is included in your manuscript, as the ethics statement entered into the online submission form will not be published alongside your manuscript. 

3. In the online submission form, you indicated that The data that support the findings of this study are available from the Health Product Vigilance Center (HPVC), and the Food and Drug Administration of Thailand (Thai FDA), but restrictions apply to the availability of these data, and so are not publicly available. Data are however available from the authors upon reasonable request and with permission of the concerned department. 

5. Please ensure that you refer to Figure 5 in your text as, if accepted, production will need this reference to link the reader to the figure.

Reviewers' comments:

Reviewer's Responses to Questions

**Comments to the Author**

1. Is the manuscript technically sound, and do the data support the conclusions?

Reviewer #1: Partly

Reviewer #2: Yes

Reviewer #3: Yes

2. Has the statistical analysis been performed appropriately and rigorously? 

Reviewer #1: Yes

Reviewer #2: Yes

Reviewer #3: Yes

3. Have the authors made all data underlying the findings in their manuscript fully available?

Reviewer #1: Yes

Reviewer #2: Yes

Reviewer #3: Yes

4. Is the manuscript presented in an intelligible fashion and written in standard English?

Reviewer #1: Yes

Reviewer #2: Yes

Reviewer #3: Yes

5. Review Comments to the Author

Reviewer #1: Thank you for the opportunity to review the manuscript ‘Epidemiology of Serious Adverse Drug Reactions Due to Anti-Infectives in Thailand' submitted for publication in the PlosOne

This is an interesting study analysing ADR reports from the national pharmacovigilance database of Thailand. Logistic regression model and CART model were used to compare serious and non-serious reported ADRs related to anti-infective drugs. The manuscript is clear and well written.

With this stated, I want to make some critical comments with the purpose of stimulating important improvements.

The main comment relates to the representativeness of the data. The author mentioned under-reporting, but an important source of bias is the lack of representativeness due to spontaneous reporting. In general, reported data do not allow conclusions to be drawn about the frequency of ADRs, but about the frequency of reporting, since reporting is not representative of the occurrence in population. This limitation needs to be discussed and all claims such as "males... were more likely to experience serious ADRs", "serious ADRs are more commun..." need to be modified. The title is also not adapted, as this is not a study on the frequency of ADRs in the population.

Authors should explain important differences between regions.

The difference between IPD and OPD patients may be due to differences in reporting in hospital versus community practice. Did ADRs occur in IPD patients during hospitalization or were patients hospitalised due to ADRs?

The authors concluded that “The beta-lactam antibiotics subgroup had a higher proportion of serious ADRs due to the anti-infective drug. The results of this study will enable healthcare professionals to use caution when prescribing to those groups” but such a conclusion should be qualified by the high proportion of beta-lactam prescriptions. Serious ADRs are generally rare, i.e. less than 1 in 10000. Then such reports only occurred with highly used drugs such as beta-lactam. If serious ADRs were proportionally less reported with other antibiotics, this could be due to differences in exposure.

The main originality of this study is the use of the CART model with multivariate logistic regression. The value of the CART model over the regression model or disproportionality analysis should be emphasised.

A minor comment concerns redosing. It is presented as a response to ADRs and not as a cause of ADRs. It is not clear to me why redosing is a response to death (Figure 3).

Reviewer #2: -Title: The title is concise and appropriately conveys the focus of the study. However, adding "A Nationwide Study" might clarify the scope.

- Abstract: The abstract effectively summarizes the study's background, methods, results, and conclusions. Consider rephrasing some sections for clarity, e.g., changing "may be fatal" to "can result in fatal outcomes."

Introduction

- The introduction provides an adequate background on adverse drug reactions (ADRs), their impact, and the importance of studying serious ADRs in Thailand.

- Suggestions: Highlight the gap in research specific to anti-infectives in Thailand earlier to emphasize the study's importance.

Results

- Clarity: Some results could be simplified, as too much numerical detail might overwhelm readers. For example, summarizing the age and region-based findings with more focus on the higher-risk groups could improve readability.

- Figures: Figures 1-6 and tables are informative but would benefit from more descriptive legends, especially in the CART model (Figures 4-6) to help interpret nodes and branches.

Discussion

- Interpretation: The discussion effectively interprets the findings and contextualizes them with existing literature. Expanding further comparison with other ADR studies globally and regionally will strengthen the analysis.

- Limitation: The manuscript mentions missing data and the potential for under-reporting in the ADR reporting system. Consider expanding on these limitations and suggesting specific improvements to address them.

- Practical Implications: Highlighting the implications for healthcare policy and the importance of ADR tracking in Thailand adds value.

This manuscript offers significant insights into ADR epidemiology, particularly for high-risk groups in Thailand. Addressing the minor clarity and structural issues will enhance its impact on readers.

Reviewer #3: 1. When writing the results in the abstract section, it is better to put the lower bound of the confidence interval first instead of vise versa, as a result it will be clearly understandable for readers. For example see the following statement "The most commonly reported serious ADRs were in the South region of Thailand (OR=1.92, 95% CI=1.97-1.88), followed by the North region (OR=1.68, 95% CI= 1.71-1.64) of Thailand."

2. Your introduction section lacks clear justification for why you did the study, in the presence of many other similar studies. Make it more clear and better to emphasize what is the novelty of your study.

3. What was your base line to classify age groups in that way?

4. In your table four, There are variables which are significant as per the confidence interval but are missed in your significant variables list. please check and include it.

5. You mentioned in the discussion section that you used retrospective study which a "strength" of your study. However with the possible attrition rates and missed data related to retrospective study, I recommend if you could mention it as a "limitation" to further recommend prospective studies.

6. PLOS authors have the option to publish the peer review history of their article (what does this mean? ). If published, this will include your full peer review and any attached files.

**Do you want your identity to be public for this peer review?** For information about this choice, including consent withdrawal, please see our Privacy Policy .

Reviewer #1: **Yes: ** Patrick MAISON

Reviewer #2: No

Reviewer #3: No

---

## [Author Response · Author response to Decision Letter 1]

26 Dec 2024

Response to Reviewer

Dear Editor,

We appreciate you and the reviewers taking the time to review our paper and provide valuable feedback. Your valuable and insightful comments prompted possible improvements to the current version. The authors carefully considered the comments and did our best to address each one. We hope that the manuscript, after careful revisions, meets your high standards. The authors welcome any additional constructive feedback.

The responses are listed below, point by point. All modifications to the manuscript have been highlighted in yellow.

Sincerely,

Dr. Khurram,

Response to Reviewer 1

Reviewer #1: Thank you for the opportunity to review the manuscript ‘Epidemiology of Serious Adverse Drug Reactions Due to Anti-Infectives in Thailand' submitted for publication in the PlosOne.

This is an interesting study analyzing ADR reports from the national pharmacovigilance database of Thailand. Logistic regression model and CART model were used to compare serious and non-serious reported ADRs related to anti-infective drugs. The manuscript is clear and well written.

With this stated, I want to make some critical comments with the purpose of stimulating important improvements.

Thank you very much for giving a positive and constructive response. We try our best to resolve the concerns and suggestions to improve the quality of our manuscript.

The main comment relates to the representativeness of the data. The author mentioned under-reporting, but an important source of bias is the lack of representativeness due to spontaneous reporting. In general, reported data do not allow conclusions to be drawn about the frequency of ADRs, but about the frequency of reporting, since reporting is not representative of the occurrence in population. This limitation needs to be discussed and all claims such as "males... were more likely to experience serious ADRs", "serious ADRs are more common..." need to be modified. The title is also not adapted, as this is not a study on the frequency of ADRs in the population.

Thank you very much for pointing this out. We revised the whole manuscript and changed the tone of many sentences. A few of these are as follows:

> [Results, Page 7, Line 168-171], [Discussion, Page 9, Lines 216-218]:

“From the reported ADRs, males (OR = 1.11, 95% CI = 1.11-1.13) and those with a prior history of drug allergy (OR = 1.22, 95% CI = 1.20-1.24) experienced more serious ADRs compared to females and those without a history of drug allergy, respectively.”.

>[Abstract, Page2, Line 40-42]: “Reported ADRs revealed that patients were males (OR = 1.11, 95% CI = 1.11-1.13) and those with a prior history of drug allergy (OR = 1.22, 95% CI = 1.20-1.24) were more likely to experience serious ADRs.”

> [Results, Page 8, Lines 189-190]: “The CART model shows that reported ADRs in IPD patients and males were most commonly associated with serious ADRs.”

We have added the limitations at the end of the discussion as follows.

“The spontaneous reporting system for ADRs likely contributes to under-reporting, which may not be the true representative of the population. Furthermore, missing data is a limitation due to retrospective studies. Therefore, a prospective study can be conducted in the future to validate the results.”

We have revised the title as follows.

“Epidemiology of Reported Serious Adverse Drug Reactions Due to Anti-Infectives using Nationwide Database of Thailand”

Authors should explain important differences between regions.

We have revised the manuscript by adding the explanation of important differences between regions as follows:

Thailand is divided into different regions based on geographical locations by local administration. Many researchers used five regions grouping as they have different environments and cultural behaviors for example:

Chokngamwong, R., & Chiu, L. S. (2008). Thailand daily rainfall and comparison with TRMM products. Journal of hydrometeorology, 9(2), 256-266.

Hitokoto, H., Takahashi, Y., & Kaewpijit, J. (2014). Happiness in Thailand: Variation between urban and rural regions. Psychologia, 57(4), 229-244.

Waqas, M., Naseem, A., Humphries, U. W., Hlaing, P. T., Shoaib, M., & Hashim, S. (2024). A comprehensive review of the impacts of climate change on agriculture in Thailand. Farming System, 100114.

[Discussion, Page 8-9, Lines 209-213]:

“The most commonly reported serious ADRs were in the south region of Thailand (OR = 1.92, 95% CI = 1.88-1.97) compared to the central region of Thailand. The possible reasons may be due to the longer rainy season in the South of Thailand which causes a higher risk of sickness, which is consistent with the study indicating that season is a factor associated with serious ADRs [28].”

The difference between IPD and OPD patients may be due to differences in reporting in hospital versus community practice. Did ADRs occur in IPD patients during hospitalization or were patients hospitalized due to ADRs?

Thank you for pointing out this concern. We already added the word “reported ADRs” and also mentioned the limitations to justify your concern.

We deleted the sentence that serious ADRs are more common in IPD patients than OPD patients.

The authors concluded that “The beta-lactam antibiotics subgroup had a higher proportion of serious ADRs due to the anti-infective drug. The results of this study will enable healthcare professionals to use caution when prescribing to those groups” but such a conclusion should be qualified by the high proportion of beta-lactam prescriptions. Serious ADRs are generally rare, i.e. less than 1 in 10000. Then such reports only occurred with highly used drugs such as beta-lactam. If serious ADRs were proportionally less reported with other antibiotics, this could be due to differences in exposure.

We agree with the reviewer that beta-lactam antibiotics were highly prescribed and that’s why reported ADRs were higher. Your suggestion is valuable if we said that the ADRs were higher due to beta-lactam but instead, we compared serious with non-serious ADRs. We can see that the percentage of serious ADRs was higher than that of non-serious ADRs. So it is not just about the number of users but the percentage of people who had serious ADRs in all reported ADRs. It can be seen from Table 2 that from all reported ADRs 42.1% ADRs were non-serious while 57.9% ADRs were serious. Complementary, Cephalosporin had 79.8% reported non-serious ADRs as compared to serious ADRs. So we can say that from reported ADRs related to Cephalosporin, the percentage of non-serious ADRs is much higher than serious ADRs.

To moderate the concern due to reported ADRs, we have written it as:

“The beta-lactam antibiotics subgroup had a higher percentage of reported serious ADRs due to the anti-infective drug.”

Instead of “The beta-lactam antibiotics subgroup had a higher proportion of serious ADRs due to the anti-infective drug. The results of this study will enable healthcare professionals to use caution when prescribing to those groups”.

The main originality of this study is the use of the CART model with multivariate logistic regression. The value of the CART model over the regression model or disproportionality analysis should be emphasized.

We are grateful for the reviewer's concern. We have focused on the values of the CART model as suggested by revising the explanation of results. We have added the following line in the conclusion.

“Furthermore, developing a reporting system to reduce serious ADR evidence, such as software with electronic prescribing databases or applications that enable efficient detection of ADRs in high-risk groups using CART model, was critical in order to closely monitor and improve patient safety.”

A minor comment concerns redosing. It is presented as a response to ADRs and not as a cause of ADRs. It is not clear to me why redosing is a response to death (Figure 3).

Redosing might be a factor that affects the result of ADRs in a worse situation like death. To make the caption of each figure to be clearer, we have revised it accordingly as follows.

[Page 22, Lines 410]: “Fig 3. Association between Redosing and Type of Anti-Infective-Induced-ADRs”

[Page 15, Lines 381]: “S3 Fig. Association between Redosing and Type of Anti-Infective-Induced-ADRs”

Response to Reviewer 2

Reviewer #2: -Title: The title is concise and appropriately conveys the focus of the study. However, adding "A Nationwide Study" might clarify the scope.

We have revised the title as follows.

“Epidemiology of Reported Serious Adverse Drug Reactions Due to Anti-Infectives using Nationwide Database of Thailand”

- Abstract: The abstract effectively summarizes the study's background, methods, results, and conclusions. Consider rephrasing some sections for clarity, e.g., changing "may be fatal" to "can result in fatal outcomes."

The term “may be fatal” is altered to “can result in fatal outcomes” as follows:

[Page 2, Lines 26]: “Serious Adverse Drug Reactions (ADRs) can cause a longer stay, which can result in fatal outcomes”

[Introduction, Page 3, Lines 57-58]: “Serious ADRs can cause a longer stay, which can result in fatal outcomes”

Introduction

- The introduction provides an adequate background on adverse drug reactions (ADRs), their impact, and the importance of studying serious ADRs in Thailand.

We thank you for the positive response.

- Suggestions: Highlight the gap in research specific to anti-infectives in Thailand earlier to emphasize the study's importance.

We presented a gap in research that the previous research applied a nationwide database in Thailand to investigate the factors associated with serious ADRs due to the Dimenhydrinate drug, but anti-infective drugs have been limited. [Page 3, Line 93-96]

[Introduction, Page 3-4, Lines 74-77]: “The previous research used Thai Vigibase to study the factors associated with serious outcomes of ADRs caused only by Dimenhydrinate, which are of limited scope. Therefore, the purpose of this study was to analyze a large-scale nationwide database in Thailand to investigate the predisposing factors associated with serious ADRs and explore drug exposure to ADRs and their pattern.”

Results

- Clarity: Some results could be simplified, as too much numerical detail might overwhelm readers. For example, summarizing the age and region-based findings with more focus on the higher-risk groups could improve readability.

In the result section, explaining other important variables, such as the Anatomical Therapeutic Drug Class (ATC) and anti-infective drug type, is considered important information to explain the culprit drug, helping to explain drug causes of serious ADRs. Moreover, this result links to the CART model result, and connections to other research are explained in the discussion section.

- Figures: Figures 1-6 and tables are informative but would benefit from more descriptive legends, especially in the CART model (Figures 4-6) to help interpret nodes and branches.

Thank you for the suggestion. If we add more details in the legend, then the legend will be unnecessary larger and make confusion for the reader and occupy more space. To address your concern, we have added the details of nodes and branches in the results section.

Discussion

- Interpretation: The discussion effectively interprets the findings and contextualizes them with existing literature. Expanding further comparison with other ADR studies globally and regionally will strengthen the analysis.

We explained further comparisons with other ADR studies globally and regionally as follows: [Page 7, Lines 219-226]

[Discussion, Page 8, Lines 201-208]: “A few studies explore the related risk factors for the seriousness of ADRs. The study conducted in South Korea [8] revealed that polypharmacy and liver function tests (AST/ALT ratio) must be monitored carefully within high-risk groups for serious ADRs. Moreover, the study conducted in China [28] found that age, number of medications and illnesses, level of medical institution, history of adverse reactions, seasons, and type and method of medication were all factors that affected serious ADRs. On the other hand, the present study's finding shows that serious ADRs are statistically associated with region, gender, ethnicity, age, type of patient, history of drug allergy, chronic disease, and dose frequency (p-value < 0.001).”

- Limitation: The manuscript mentions missing data and the potential for under-reporting in the ADR reporting system. Consider expanding on these limitations and suggesting specific improvements to address them.

We explained more on under-reporting and suggestions as follows: [Page 8, Lines 260-263]

[Discussion, Page 10, Lines 254-256]: “The spontaneous reporting system for ADRs likely contributes to under-reporting, which may not be the true representative of population. Furthermore, missing data is a limitation due to retrospective studies. Therefore, a prospective study can be conducted in the future to validate the results.”

- Practical Implications: Highlighting the implications for healthcare policy and the importance of ADR tracking in Thailand adds value.

We explained more on practical implications as follows:

[Discussion, Page 10, Lines 243-247]: “Furthermore, this study analyzed a large-scale nationwide database in Thailand using LR and CART analysis methods to investigate risk factors associated with serious ADRs that have an impact on health outcomes and the cost of patient hospital care. Policy makers have the influence and opportunity to use the research evidence to alter or develop effective policies in order to prevent serious ADRs and the health authorities can make decisions more quickly to restrict a drug’s use or pay close attention to high risk group for patient safety.”

This manuscript offers significant insights into ADR epidemiology, particularly for high-risk groups in Thailand. Addressing the minor clarity and structural issues will enhance its impact on readers.

Thank you very much for your positive comments.

Response to Reviewer 3

Reviewer #3: 1. When writing the results in the abstract section, it is better to put the lower bound of the confidence interval first instead of vise versa, as a result it will be clearly understandable for readers. For example see the following statement "The most commonly reported serious ADRs were in the South region of Thailand (OR=1.92, 95% CI=1.97-1.88), followed by the North region (OR=1.68, 95% CI= 1.71-1.64) of Thailand."

We have revised all of the 95%CI values in the manuscript by putting lower bound first as suggested.

2. Your introduction section lacks clear justification for why you did the study, in the presence of many other similar studies. Make it more clear and better to emphasize what is the novelty of your study.

We presented a gap in research that the previous resear

---

## [Decision Letter · Decision Letter 1]

20 Jan 2025

Epidemiology of Reported Serious Adverse Drug Reactions Due to Anti-Infectives Using Nationwide Database of Thailand

PONE-D-24-43299R1

Dear Dr. Khuram,

We’re pleased to inform you that your manuscript has been judged scientifically suitable for publication and will be formally accepted for publication once it meets all outstanding technical requirements.

Kind regards,

Obed Kwabena Offe Amponsah, PharmD, Ph.D.

Academic Editor

PLOS ONE

Additional Editor Comments (optional):

Reviewers' comments:

Reviewer's Responses to Questions

**Comments to the Author**

1. If the authors have adequately addressed your comments raised in a previous round of review and you feel that this manuscript is now acceptable for publication, you may indicate that here to bypass the “Comments to the Author” section, enter your conflict of interest statement in the “Confidential to Editor” section, and submit your "Accept" recommendation.

Reviewer #1: All comments have been addressed

Reviewer #2: All comments have been addressed

Reviewer #3: All comments have been addressed

2. Is the manuscript technically sound, and do the data support the conclusions?

Reviewer #1: Yes

Reviewer #2: Yes

Reviewer #3: (No Response)

3. Has the statistical analysis been performed appropriately and rigorously? 

Reviewer #1: Yes

Reviewer #2: Yes

Reviewer #3: (No Response)

4. Have the authors made all data underlying the findings in their manuscript fully available?

Reviewer #1: Yes

Reviewer #2: Yes

Reviewer #3: (No Response)

5. Is the manuscript presented in an intelligible fashion and written in standard English?

Reviewer #1: Yes

Reviewer #2: Yes

Reviewer #3: (No Response)

6. Review Comments to the Author

Reviewer #1: (No Response)

Reviewer #2: (No Response)

Reviewer #3: (No Response)

7. PLOS authors have the option to publish the peer review history of their article (what does this mean? ). If published, this will include your full peer review and any attached files.

**Do you want your identity to be public for this peer review?** For information about this choice, including consent withdrawal, please see our Privacy Policy .

Reviewer #1: **Yes: ** Patrick Maison

Reviewer #2: No

Reviewer #3: No

---

## [Editor Report · Acceptance letter]

PONE-D-24-43299R1

PLOS ONE

Dear Dr. Khurram,

I'm pleased to inform you that your manuscript has been deemed suitable for publication in PLOS ONE. Congratulations! Your manuscript is now being handed over to our production team.

Kind regards,

on behalf of

Dr. Obed Kwabena Offe Amponsah

Academic Editor

PLOS ONE